# Gene Augmentation and Editing to Improve TCR Engineered T Cell Therapy against Solid Tumors

**DOI:** 10.3390/vaccines8040733

**Published:** 2020-12-03

**Authors:** Vania Lo Presti, Frank Buitenwerf, Niek P. van Til, Stefan Nierkens

**Affiliations:** 1Center for Translational Immunology, University Medical Center Utrecht, 3584 CX Utrecht, The Netherlands; vloprest@umcutrecht.nl (V.L.P.); f.buitenwerf@gmail.com (F.B.); snierken@umcutrecht.nl (S.N.); 2AVROBIO, Cambridge, MA 02139, USA; 3Princess Máxima Center for Pediatric Oncology, 3584 CS Utrecht, The Netherlands

**Keywords:** adoptive T cell therapy, T cell receptor engineering, gene editing, solid tumors

## Abstract

Recent developments in gene engineering technologies have drastically improved the therapeutic treatment options for cancer patients. The use of effective chimeric antigen receptor T (CAR-T) cells and recombinant T cell receptor engineered T (rTCR-T) cells has entered the clinic for treatment of hematological malignancies with promising results. However, further fine-tuning, to improve functionality and safety, is necessary to apply these strategies for the treatment of solid tumors. The immunosuppressive microenvironment, the surrounding stroma, and the tumor heterogeneity often results in poor T cell reactivity, functionality, and a diminished infiltration rates, hampering the efficacy of the treatment. The focus of this review is on recent advances in rTCR-T cell therapy, to improve both functionality and safety, for potential treatment of solid tumors and provides an overview of ongoing clinical trials. Besides selection of the appropriate tumor associated antigen, efficient delivery of an optimized recombinant TCR transgene into the T cells, in combination with gene editing techniques eliminating the endogenous TCR expression and disrupting specific inhibitory pathways could improve adoptively transferred T cells. Armoring the rTCR-T cells with specific cytokines and/or chemokines and their receptors, or targeting the tumor stroma, can increase the infiltration rate of the immune cells within the solid tumors. On the other hand, clinical “off-tumor/on-target” toxicities are still a major potential risk and can lead to severe adverse events. Incorporation of safety switches in rTCR-T cells can guarantee additional safety. Recent clinical trials provide encouraging data and emphasize the relevance of gene therapy and gene editing tools for potential treatment of solid tumors.

## 1. Advantages and Disadvantages of Using rTCR-T Cells for the Treatment of Solid Tumors: From the Bench to the Bedside 

The endogenous immune system is a key regulator in recognizing and controlling tumor growth. However, many cancers are able to modulate the anti-tumor properties of immune cells by preventing recognition by immune cells, counteracting the influx of cells, or actively suppressing their function. Cancer immunotherapy has been one of the most exciting approaches in the fight against cancer over the past several decades. The aim of this type of therapy is to strengthen the patients’ immune system to recognize, target, and subsequently eliminate tumor cells [1]. For instance, clinical successes have been obtained by monoclonal antibodies functioning as immune checkpoint inhibitors to increase the anti-tumor effects of T lymphocytes. In 2011, the first checkpoint inhibitor targeting cytotoxic T lymphocyte-associated protein 4 (CTLA-4) was approved by the FDA (Ipilimumab), followed by several other drugs targeting CTLA-4, programmed cell death protein 1 (PD-1) and PD-1 ligand (PD-L1) [2]. However, even if application of checkpoint inhibitors may be durable in some cases and lead to remission of tumor growth, the response rates to date have rarely exceeded 40% [3]. 

An alternative strategy that has emerged to strengthen the immune system for cancer treatment is adoptive cell transfer (ACT). This technique is based on selection of immune cells to enhance their ability to recognize and eliminate tumor cells [4]. ACT has proven to be effective in several cancer types, especially using the transfer of T lymphocytes [5]. Moreover, gene modification techniques have increased the success rate of ACT against cancer by transferring a specific receptor, either a chimeric antigen receptor (CAR) or a T cell receptor (TCR), able to recognize a tumor associated antigen (TAA). Treatment of hematological tumors has been particularly successful, leading to the first approval of ACT using CD19-CAR (CAR19) T cells for the treatment of acute lymphoblastic leukemia (ALL) in 2017 (Kymriah) [6]. Although many hematological cancer patients have received clinical benefits from CAR-T cell treatment, so far, it has been less effective in the treatment of solid tumors. Several researchers have recently summarized the latest clinical trials involving the use of CAR-T cells in solid tumor treatment, highlighting the potency and limitations of the approach [7,8]. Solid tumors comprise an immunosuppressive tumor microenvironment characterized by a dense tumor stroma and by the presence of immunosuppressive signals, creating both a physical and chemical barrier for immune cells to infiltrate and to act [9]. Moreover, solid tumors are characterized by a heterogeneous antigen expression, which is in part responsible for the treatment response failure [10]. The identification of an ideal target antigen is a limiting factor for two principal reasons. First, tumor associated antigens (TAA) are often expressed on healthy tissue and cells, raising safety concerns. Second, tumors have a low percentage of cell surface TAA expression (1 to 10% depending on the tumor type). For the latter problem, rTCR-T cells might have advantages for the solid tumor treatment [11].

rTCR-T cells contain TCRs recognizing a specific TAA when presented by a human leukocyte antigen (HLA) molecule. The majority of T cells typically express a TCR comprising of an α and β chain, which can interact with peptides presented on the HLA molecules in association with co-receptors (e.g., CD4 and CD8) [12,13,14]. rTCR-T cells are restricted to HLA recognition but recognize both extracellular and intracellular proteins, a major advantage compared to CAR T cells, since more than 85% of cellular proteins are located intracellularly (Figure 1) [15].

However, the advantage of recognizing peptides bound to the HLA molecules also increases the possibility to recognize a non-cognate peptide, derived from a different protein, the so-called off-target effect. Unfortunately, this led to an unpredictable outcome in clinical trials, with patients developing strong inflammatory responses [16] and in some cases resulting in patients’ death [17,18]. As a result of advanced screening procedures for on-target and off-target side effects, rTCR-T cells with low toxicity have been implemented in recent clinical trials. A list of clinical trials targeting specific antigens for the treatment of solid tumors can be found in Table 1. In cases where solid tumors failed to respond to ACT, complementary treatments to improve efficacy and safety of the adoptively transferred T cells, such as prior lymphodepletion, interleukin 2 (Aldesleukin; IL-2), and PD-1 blockade have been incorporated. Both lymphodepletion and IL-2 administration have been shown to promote survival and proliferation of the infused engineered T cells [19], whereas PD-1 blockade decreases T cell exhaustion [20]. Even if rTCR-T cells can have an advantage over other therapies in the context of solid tumors, improvements are still necessary to increase efficacy and safety. This review summarizes the most recent developments, with a strong focus on genetic approaches, that have been applied in the field of rTCR-T cells and the treatment of solid tumors.

## 2. Increasing Affinity and Functional Avidity of rTCRs While Maintaining a Safe Profile

To obtain an efficacious response by rTCR-T cell therapy, it is necessary to select TCRs with high functional avidity and affinity to TAA. The identification of tumor infiltrating lymphocytes (TILs) often leads to the isolation of potent TCRs. All TCRs used in clinical trials are derived from T cells isolated from cancer patients or from allogeneic donor cells [21,22,23]. Genetic modification of key amino acids, identified with the use of phage display and point mutations, has shown the possibility to further increase the TCR affinity, evaluated measuring binding of the recombinant TCRs (rTCRs) to the HLA-bound cognate peptides [24,25]. If the TCR affinity is important at the level of antigen recognition, TCR functional avidity is essential to obtain an actual cytotoxic effect. Enhancing the rTCR expression on the cell surface is one of the strategies to increase the functional avidity. A high level of rTCR expressed on the cell surface can double the production of cytotoxic cytokines, such of IFNγ, compared to a low expression [26]. Retroviral and lentiviral vectors are the golden standard for delivering rTCRs genes into primary T cells. Over the years, mutations in the envelope of the vectors [27] and culture systems [28,29] have been optimized to specifically increase the transduction efficiency of T cells and optimize a stable expression of the rTCR on the cell surface. Moreover, mutations in the TCR constant regions, such as the introduction of additional cysteine di-sulfide bonds, sequences of murine origin, or genetic bond to the CD3ζ-complex, have been used to increase rTCR expression and reduce human TCR α and β chain mispairing with endogenous TCRs, but often resulted in a partial success [30,31]. More recently, it has been demonstrated that substitutions of 3 amino acid residues in the TCR variable domains consistently increase the expression of rTCRs on the surface of engineered T cells and the efficacy of the final product without acting on its affinity [32]. Intrinsic inhibitory signaling can also play an important role in the expression and functional avidity of the TCR on the cell surface. For instance, the cytokine-inducible SH2-containing (CISH) protein physically interacts with the TCR intermediate phospholipase Cγ1 (PLC-γ1), targeting it for proteasomal degradation after TCR stimulation by the cognate peptide. The depletion of CISH unleashed a TCR-dependent hyperactive program, resulting in the upregulation of pro-functional, proliferative, and survival genes. Moreover, the genetic deletion of CISH, in a mouse model of melanoma, significantly increases CD8^+^ T cells related cytokines production (IFNγ, TNFα, and IL-2) and anti-tumor reactivity, improving the survival of mice for more than 60 days. The effect of CISH downregulation was also evaluated using peripheral blood CD8^+^ T cells expressing a rTCRs. CD8^+^ T cells, treated with CISH silencing RNA (siRNA), showed a 2 fold increase in cytokine production when co-cultured with target tumor cells [33]. Recently, a Phase I/II clinical trial on metastatic gastrointestinal cancers has started the patients recruiting phase to evaluate the effect of TILs in which the CISH gene is inhibited using gene editing techniques (NCT03538613). Increasing affinity and avidity of the TCRs have a reflection on the safety of the T cell therapy. This has been demonstrated in clinical trials with a high affinity rTCR-T cells against melanoma antigen recognized by T cells 1 (MART-1) causing severe toxicities of the skin, eye, and ears [34]. On the contrary, in an earlier clinical trial including melanoma patients, low toxicity and respectable toleration were observed when using a TCR with lower affinity for MART-1 [35]. In addition, to preserve the safety of the therapy, an extensive analysis of the expression of the antigen on healthy tissue and the possible cross-reactivity against non-cognate peptide is essential before using rTCR in the clinic. In particular, the recognition of key amino acid residues allows the prediction of possible cross-reactive antigens. Bijen et al. tested the cross-reactivity of a histocompatibility antigen 2 (HA2) specific TCR using a 9-mer combinatorial peptide library (CPL) screening. This technique was able to recognize cross-reactivity toward a Cadherin 13 (CDH13)-derived peptide, not detected using the most frequent test (alanine scanning mutagenesis). Further experiments proved the ability from the HA2 specific rTCR-T cells to recognize healthy cells, such as fibroblasts and keratinocytes, known as CDH13 expressing cells [36]. Several techniques to better predict cross-reactivity of rTCRs have been explored in the last few years, extensively illustrated in a recent review by Bentzen and Hadrup [37]. The cross-reactivity prediction tools in vitro help minimize the onset of unexpected serious side effects that have been reported in previous clinical trials. In 2013, two clinical trials reported unexpected toxicities derived from the administration of melanoma associated antigen (MAGE)-A3-rTCR-T cells in melanoma patients. The first trial showed serious neurological reaction in 3 out of 9 patients treated. Post mortem evaluation of patients’ brain showed positivity for MAGE-A12, one of the recognized epitopes by the TCR used in this study [17]. The other clinical trial reported that two treated patients developed cardiogenic shock and died within a few days upon the MAGE-A3-rTCR-T cells. Only afterwards the rTCR-T cells were found to be cross-reactive to a similar epitope derived from the striated muscle-specific protein titin, expressed by cardiac cells [18]. 

## 3. Genetic Elimination of Endogenous TCR to Improve Efficacy and Safety 

Preferably, rTCR-T cells intended for ACT should only express the rTCR α and β chain of interest. However, conventional gene modification techniques often use viral systems to introduce the rTCR, without eliminating the expression of the endogenous TCR (eTCR). The presence of the eTCR α and β chains can lead to mispairing with the rTCR subunits [38], increasing the chances of creating novel peptide recognition that, as shown in a lymphopenic mouse model, can lead to lethal graft versus host disease [39]. Additionally, TCR heterodimer mispairing reduces the formation of the correct rTCRs and compete in the formation of complexes with co-receptors, reducing the functionality of the T cell product [40]. As discussed in the previous paragraph, molecular techniques can be used on the sequence of the rTCR to decrease mispairing with the eTCR. On the other hand, genome editing techniques have been developed to completely eliminate the expression of the eTCR targeting the constant region of TCR α and β chains. Zinc-finger nucleases (ZFN) and meganucleases have initially been used, but these are cumbersome to develop, and have resulted in only 7% of gene-editing frequencies in primary T cells [41]. More recently, transcription activator-like effector nuclease (TALENs) and clustered regularly interspaced short palindromic repeats (CRISPR)/Cas9 endonuclease technologies have been developed, which are more attractive due to an easier design and high efficiency gene-editing results. As recently summarized by Zhang et al., the use of CRISPR/Cas9 technology in primary T cells increased the success rate of gene editing compared to ZFN and TALENs [42] and has, in some cases, reached up to 90% of target gene deletion [43]. With these techniques, gene editing multiple genes at the same time (multiplexing) is feasible; for example, CRISPR/Cas9 multiple guide RNAs (gRNAs) can be delivered simultaneously [43,44,45,46]. In 2017, a pioneer clinical administration of CAR19 T cells, gene edited with TALEN to simultaneously deplete eTCR and CD52, a target of the serotherapeutic molecule alemtuzumab, showed the successful induction of molecular tumor remission ahead of allogeneic stem cell transplantation in two pediatric B-ALL patients. Additionally to the beneficial effects of the eTCR disruption, the genetic disruption of CD52 expression protected the infused cells from the depleting effect of alemtuzumab. Analysis of gene edited T cells, pre-infusion, showed expression of CAR19 in 85% of cells and depletion of both eTCR α and β chains and CD52 in more than 64% of the cells [45]. More recently, a first-in-human phase I clinical trial has started to test the safety and efficacy of rTCR-T cells, in which the constant regions of eTCR α and β chains (*TRAC* and *TRBC*) and PD-1 genes were knocked out (KO) using CRISPR/Cas9. The frequency of editing varied according to the gRNAs and was approximately 45% for TRAC, 15% for TRBC, and 20% for PD-1. It was demonstrated that this multiplexing approach, in conjunction with lentiviral delivery of a TCR recognizing an epitope of NY-ESO-1 and LAGE-1 cancer testis antigens, was feasible to generate triple knockout T cells with recombinant TCR and with initial favorable safety profiling. In one patient analyzed at depth, a frequency of 30% of di-genic and tri-genic editing was achieved in the infused cell population. Infusion of the T cell product in two patients with advanced refractory myeloma and one with metastatic sarcoma demonstrated an on-target effect, with no clinical toxicities. After 100 days, two out of the three patients had a stable disease, cytokine release syndrome did not occur and neither did any other infusion related side effects. Unfortunately, tumor progression was observed after 300 days in all patients, of which two received additional salvage chemotherapy agents and one died from advanced stage myeloma [47]. One of the main challenges of this multistep and multiplex approach is to generate the therapeutic product under GMP compliant conditions. Therefore, reducing the complexity of the process by limiting the number of the necessary components should be beneficial for large-scale clinical transition. A relevant approach is to disrupt the eTCR locus while simultaneously integrating the rTCR in the TCR locus of T cells using CRISPR/Cas9 mediated homologous recombination (HR). Adeno associated viral (AAV) vectors have been used as DNA donor sequence delivery system, especially serotype AAV6 [48], particularly efficient to provide HR compared to other viral systems, such as integration deficient lentiviral vectors [49]. The use of AAV6 and HR has been reported for the successful generation of CAR19 T cells. Three studies published in 2017 showed a targeted integration of the CAR19 gene in the *TRAC* locus in 38% to 45% of primary T cells, inducing a simultaneous loss of TCR expression and proficient expression of the CAR. The administration of *TRAC^KO^*-CAR19 T cells in the tumor mice model showed greater responses and prolonged median survival at every T-cell dose compared to only transduced CAR19 T cells [50,51,52]. Other groups evaluated the possibility to deliver the HR donor sequence without the use of viral vectors to avoid immunogenicity, due to the presence of AAV derived proteins that can activate the pre-existing humoral response in vivo [53], and to accelerate clinical application, considering that non-viral materials typically can be more easily adapted to good manufacturing practices for clinical use. Roth et al. were the first to describe a protocol in which the electroporation of the CRISPR/Cas9 ribonucleoprotein complex (RNP) and a dsDNA template coding for an NY-ESO specific TCR was successfully integrated in 12% of target T cells via HR [54]. Furthermore, major advantages of a targeted integration are that the transcriptional regulation would be physiological, under the control of the endogenous promoter, and the risk of vector-induced insertional mutagenesis is minimized. In a later study, Schober et al. demonstrated the importance of simultaneously targeting both *TRAC* and *TRBC* genes, as only knocking in the rTCR into the *TRAC* leads to an even increased mispairing between the recombinant α-chain and endogenous β-chain, compared to TRAC^KO^ viral transduced rTCR-T cells and not gene-edited viral transduced rTCR-T cells. Notably, inserting the desirable rTCR in the *TRAC* locus while concurrently knocking out the *TRBC* gene leads to a harmonized expression of the recombinant TCR on the cell surface (Figure 2), ultimately increasing the efficiency of the response against tumor cells in vitro, with an increased production of IFNγ upon antigen recognition [55]. In conclusion, disruption of both TCR α and β genes can diminish mispairing and can thereby increase efficacy and safety with respect to potential off-target autoimmunity.

Clinical benefits have yet to be elucidated in more human clinical trials to provide solid evidence of this technique to improve rTCR-T cell therapy against solid tumors. 

## 4. Disrupting Inhibitory Pathways to Prevent Exhaustion 

Immune checkpoint receptors on infused rTCR-T cells are differentially expressed compared to naturally circulating T lymphocytes, and that exhaustion markers are often rapidly upregulated after infusion in vivo [56]. There are many checkpoint receptors, but PD-1, LAG-3, and TIM-3 are commonly modulated by the tumor microenvironment to lead to exhaustion, endogenous T cells as well as gene modified cells [57,58]. The inhibitory immune checkpoint molecule PD-1 has been reported to be overexpressed in rTCR-T cells, especially after infusion, resulting in a diminished IFN-γ production and thereby a decreased immune response [56]. Hence, it was hypothesized that the disruption of the PD-1/PDL-1 axis could lead to less T cell exhaustion and improved persistency, and thereby an enhanced immune response. It has been shown that the use of anti-PD1 antibody augments the efficacy of NY-ESO engineered T cells both in vitro and in vivo model of human lung cancer. Repeated intraperitoneal injection of anti-PD1 antibody, perhaps, was able to halve the tumor growth compared to the injection of only rTCR-T cells [59]. Two clinical trials are now recruiting to test the effect of this combination in patients (NCT03578406; NCT04139057), in which it was already reported that two out of four treated patients displayed evidence of tumor regression. Especially in the context of solid tumor, a pre-clinical study, in a mouse model of pleural mesothelioma, showed that the administration of either PD-1 antibody checkpoint blockade, cell-intrinsic PD-1 shRNA blockade, or a PD-1 dominant negative receptor together with CAR T-cells drastically enhanced tumor burden control and prolonged median survival [60]. The increased general availability of gene engineering techniques has also skewed the focus on knocking-out the checkpoint receptors genes, such as PD-1, resulting in permanent deletion of checkpoint inhibitory signaling. Pre-clinical studies have showed how the knock-out (KO) of PD-1 can increase the efficiency and response persistency of TILs and CAR-T cells in solid tumor settings, such as mouse models of glioblastoma and fibrosarcoma [61,62,63,64]. TILs and CAR-T cells with additional KO of PD-1 have been registered for many clinical trials (summarized by Mc Gowan et al. [65]); however, to the best of our knowledge, there are still no pre-clinical and clinical studies testing the benefits of PD-1 KO rTCR-T cells in solid tumor clearance; however, recently, a clinical trial including two myeloma patients and one sarcoma patient has shown promising results. PD-1 knockout NY-ESO rTCR-T cells appeared to have a half-life of 83.9 days, the average of the three treated subjects, compared to the half-life of seven days registered in previous clinical trials with NY-ESO rTCR-T cells without the KO [47,66]. Concurrently targeting multiple immune checkpoint receptors could be key in optimizing rTCR-T cell function, and it has already been demonstrated to be possible in pre-clinical studies in CAR T cells, with the one-shot generation of a dual inhibitory resistant universal CAR T cells deficient for TCR, HLA-I, PD-1, and CTLA-4 [43]. Specific KO of checkpoint receptors in the transferred cells can be a strategy to decrease the toxicity derived from the use of systemic monoclonal antibodies; however, suppressing the inhibitory pathways that induce exhaustion and cell death can increase the possibility of generating over-activated T cells with enhanced autoimmunity that cannot be controlled. Interestingly, Stadtmauer et al. showed that the percentage of cells with edits in the PD-1 locus decreased to ~5% of the cells expressing the transgenic TCR at four months after infusion, giving a positive note to the safe use of this strategy in humans [47]. However, to assess if this approach has a preferred safety profile specific attention, would be required in a long-term follow-up.

## 5. Risks of Using Gene-Editing Techniques

The use of gene editing techniques can raise concerns about the safety of rTCR-T cells, such as immunity against Cas9 protein, on target and off-target effects (reviewed by Ghosh et al. [67]). A major potential consequence of using gene targeting techniques is off-target double strand breaks, the recognition by the gene editing machinery of a similar genomic sequence. Cleavage at off-target sites can result in chromosomal rearrangements, including insertion, deletion, translocations, and disruption of important genes and genotoxicity [68]. Genome wide detection techniques, such as Digenome–seq, SITE–seq, and CIRCLE–seq, have been developed to understand the extent of off-target events in an unbiased way [69]. Furthermore, several approaches are now being used to minimize the off-target effect, especially focusing on the CRISPR/Cas9 technology, with optimization of the gRNAs design and the Cas proteins [70]. Point mutations in the sequence of the Cas9 protein have dramatically increased the specificity of the system and generated the so-called high fidelity Cas9 proteins (HiFi Cas9) [71,72]. Outstanding results were obtained with a particular point mutation R691A that led to reduced low off-target effect, down to 1%, while maintaining an efficient on-target effect, which was not always the case for other HiFiCas9 proteins tested [73]. Remarkably, in the three reports of the clinical administration of gene edited T cells (NCT03399448, NCT02793856) [45], off-target events were detected on a relatively low percentage of cells, which decreased over time, suggesting that these translocations did not confer a growth advantage over the other infused gene modified T cells. In the study evaluating the safety and feasibility of PD-1 gene editing autologous T cells using CRISPR/Cas9, 18 predicted off-target sites were analyzed with next generation sequencing (NGS) before the two cycles of T cells infusion. The median mutation frequency of all off-target sites was 0.05%. To further minimize the bias of analyzing only a small number of off target sites, seven samples were also evaluated using whole genome sequencing (WGS) at 100× coverage. Using the prediction tool Cas-OFFinder, not limited by the amount of variety in the protospacer-adjacent motif (PAM), mismatches, and gRNA length; 2086 potential off-target sites were predicted. Indel events were not detected within 15 base pairs (bp) up- and downstream of the sites. When each site was broadened to 200 bp up- and downstream, 84 indel events in 53 sites were detected, but all of them comprised 1-bp length variances on nucleotide repeats and are therefore not considered to be true off-target events [74]. Multiplexing approaches, in which multiple genes are targeted simultaneously, further increase the probability of genomic alterations, more specifically of large chromosomal translocations. In detail, in the clinical report from Qasim et al., using a dual-gene editing strategy with TALENs, chromosomal rearrangements were observed in 4% of infused T cells [45]. Stadtmauer et al. published in their clinical report the development of a qualified qPCR assay to assess the 12 potential translocations that could occur with the simultaneous editing of four loci: TRAC, TRBC1, TRBC2, and PD1. The TRBC1:TRBC2 chromosomal rearrangement was the most frequent, leading to a 9.3 kB deletion. Overall, the percentage of T cells with translocations was less than 10% of cells in all the three patients before infusion, and at days 30, 150, and 170 chromosomal translocations were at the limits of detection [47]. Furthermore, the most recent development of the CRISPR/Cas9 base editor technique, which allows for a targeted base substitution instead of a double strand break, can further minimize safety concerns [75]. To support the generation of T cell products with a multiplexing approach, a recent study showed that multiplex base edited T cells exhibit improved expansion and lack double strand break-induced translocations observed in T cells edited with Cas9 nuclease [76]. 

## 6. Incorporating Cytokines to Enhance T Cell Proliferation

Cytokines are immunomodulatory molecules that influence proliferation of B and T lymphocytes and have been applied in several cancer immunotherapeutic treatments. Cytokines of the γ-chain family, including IL-2, are important for T cell memory and proliferation. These cytokines exert many functions on T cells, such as enhanced proliferation, persistence, and improved antigen recognition [77]. In ACT, high dose of IL-2 is intravenously injected in concurrence with rTCR-T cells, in order to support their functionality and proliferation, but often results in multi-organ toxicities for patients [78]. Local delivery and the generation of mutated protein increased the specificity of the therapy and reduced systemic toxicity [79]. Moreover, expressing the IL-2 transgene simultaneously with the rTCR could further improve the engineered T cells with reduced systemic toxicity. In support of this, in a study in 2001, it was shown that complementary IL-2 expression could remarkably enhance melanoma specific survival of CD8^+^ T cells in vitro [80]. This was also supported by a later study, showing that IL-2 gene modified TILs could secrete sufficient amounts of IL-2 to prolong their own in vitro survival for up to six months. However, in vivo results were less encouraging. Although the IL-2 transduced T cells were well-tolerated, no significant response rates were observed in this clinical trial compared to non-IL-2-transduced T lymphocytes. The reason for the poor in vivo response in this application has not been elucidated and requires further studies [81]. Gene addition of other cytokines, such as IL-12 and IL-18, has been investigated, in order to increase functionality and maturation of the T cells. rTCR-T cells specific for gp100 with inducible constructs of IL-12 (iIL-12) and IL-18 (iIL-18) have shown elevated IFNy production after encountering their specific antigen and can therefore enhance the anti-tumor response. Administration of iIL-18 rTCR-T cells to mice bearing gp100 positive melanoma tumor cells resulted in a more persistent anti-tumor response with no detectable toxicity. Administration of iIL-12 rTCR-T cells has been shown to result in increased levels of IFNy and TNFα. However, under control of a nuclear factor of activated T cell (NFAT) sensitive promoter, iIL-12 correlated with compromised T cell persistence, enhanced plasma levels of inflammatory cytokines, and decreased survival in the gp100+ mouse model of melanoma [82]. In a later study, the use of an inducible Tet-On promoter, transiently activated by the antibiotic doxycycline, showed the same benefits in tumor control, in a mouse model of melanoma, obtained with iIL-18 without showing toxicities. Transient expression of iIL-12 showed to be sufficient to inhibit the growth of B16F10 melanoma tumors and to increase the number of tumor-infiltrating rTCR-T cells, without showing toxicity compared with the use of the NFAT-promoter [83]. Additionally, another means of reducing the systemic toxicity of IL-12 could be anchoring iIL-12 to the plasma membrane. In a recent study of tumor mouse models, the toxicity of IL-12 was reduced significantly when applying this approach [84]. IL-12 genetically engineered TILs have been tested in a clinical trial with patients suffering from metastatic melanoma. Administration of doses between 0.3 and 3 × 10^9^ cells showed an objective clinical response in 63% of the treated patients. The administered IL-12 gene engineered TILs were found in circulation for less than a month. More importantly, increasing cell doses was associated with high serum levels of IL-12 and IFNγ as well as clinical toxicities, including liver dysfunction, high fevers, and sporadic life-threatening hemodynamic instability [85]. A phase I clinical trial protocol was also published for recruiting patients with recurrent ovarian cancer and defined a safe and efficient dose of CAR-T cells gene engineered to over express IL-12 [86]. 

## 7. Introduction of Chemokines Receptors to Promote Migration and Infiltration

Migration, infiltration, and homing of T cells into solid tumors are often hampered by the presence of a suppressive environment. This poses a major problem, since the absence of T lymphocytes within most solid tumors correlates with a worse prognosis. Additionally, patients with a higher infiltration rate of immune cells often have a better prognosis. Post infusion tracking of T cells in animal models, with biomedical imaging techniques, has shown localization of the infused T cells primarily in the liver, spleen, and lungs, often more prominent than in the tumor site [87,88]. These preclinical models have underlined that migration and localization to the tumor remains a challenge for gene engineered T cells in the treatment of solid tumors. Chemokines play a major role in the migration and homing of T cells, and are also important for T cell survival and proliferation. There are many chemokines and chemokine receptors axes involved in tumor immunology and T cell responses, comprehensively reviewed by Nagarsheth et al., that have been explored in order to further enhance the effectiveness of ACT [89]. For example, cancers such as metastatic melanoma and ovarian cancer have been associated with proficient homing of tumor infiltrating T cells (TILs), due to a high production of pro-inflammatory chemokines, such as CXCL9 and CXCL10, recognized by the TILs [90]. However, other solid tumor types create a suppressive tumor microenvironment (TME) that express pro-tumoral chemokines to induce metastasis and recruit suppressive immune cells, such as the described effect of CCL12 in recruiting Treg cells [91]. Naturally, T cells only express a restricted set of chemokine receptors on their cell surface limiting their ability to always be attracted by chemokines expressed by tumor cells. Therefore, it was suggested that incorporation of chemokine receptors within T cells could possibly augment recruitment and trafficking to the tumor site. For instance, it was observed that many melanomas had a high production of the chemokine CXCL1, but that its receptor CXCR2 was not profoundly expressed on T lymphocytes. It is noteworthy that CXCL1 was also not expressed by T cells. As a proof of concept, Kershaw et al. were among the first ones to demonstrate that the introduction of CXCR2, via retroviral vector transduction of peripheral blood derived T cells, was a feasible technique to redirect the cells towards the tumor, resulting in proper IFN-γ response when activated by their cognate chemokine [92]. In the context of rTCR-T cells, it was found that co-expression of CXCR2 with MAGE-A3 specific TCRs could significantly enhance migration to the tumor site in a mice model of melanoma, resulting in reduced tumor growth [93]. Moreover, the introduction of CXCR2 receptor in CAR-T cells targeting CD70 greatly improved the tumor control and enhanced in pre-clinical models of aggressive tumors such as glioblastoma, ovarian, and pancreatic cancer [94]. This pre-clinical data demonstrate that exploiting chemokines is feasible to improve recruitment of engineered T cells to the tumor-site in a large spectrum of solid tumors. To translate this application to the clinic, a Phase I/II clinical trial is testing the safety and efficiency of TILs transduced with CXCR2 in treating patients with stage III melanoma and metastatic melanoma (NCT01740557).

## 8. Targeting the Tumor Surrounding Stroma

The abovementioned strategies focus on improving the functionality, migration, and persistence of rTCR-T cells by additional modification in the genome of the T cells. However, there is the possibility to switch the focus on the tumor itself and its microenvironment. This can be of particular interest to further personalize new therapeutic approaches based on the patient tumor characteristic.

Infiltrating T lymphocytes often accumulate in the stroma surrounding the tumor, comprising non-malignant cells and extracellular matrix (ECM). Cancer associated fibroblasts (CAFs) play an important role in tumor support by secreting pro-tumoral cytokines and by producing an excessive amount of ECM [95]. However, CAFs frequently express fibroblast activation protein alpha (FAPα), making them distinguishable from healthy cells and a potential target for cancer treatment [96]. It has been shown in a mouse model of lung cancer that treatment with FAPα-specific CAR T cells in concurrence with T cells targeting a tumor specific TAA, namely ephrin type-A receptor 2, could significantly enhance the anti-tumor effect compared to separate administration of these gene-engineered T cells [97]. 

Targeting the tumor vasculature could be an additional way to enhance tumor infiltration of rTCR-T cells in solid tumors. The abnormal tumor vasculature creates a hypoxic and acidic environment, and impedes the infiltration and function of anti-tumoral immune cells including T cells [98]. Additionally, tumor endothelial cells (ECs) have a strong suppressive function on T cells, and can directly block T cells to enter the tumor through downregulation of adhesion molecules, upregulation of inhibitory receptors, and production of FasL [99]. By adapting the tumor vasculature, a path could be paved for T cells to enter. A possible targetable molecule in this approach would be vascular endothelial growth factor receptor 2 (VEGFR-2), constitutively expressed by the tumoral endothelial cells. The use of CAR T cells targeting VEGFR-2 increases infiltration of T cells into the tumor, by means of destroying the endothelial cells in the stroma. However, the overall induced anti-tumor effect was only modest [100]. Concurrent administration of TAA specific rTCR-T cells, for gp100 or TRP-1, with VEGFR-2 specific T cells appeared to increase the anti-tumor effect in vivo significantly compared to administration of either alone. A strong persistence and increased infiltration was seen of the adoptively transferred T cells resulting in increased anti-tumor efficacy [101]. These examples demonstrated that combined therapies, targeting both the tumor microenvironment and the tumor cells, are a promising approach to improve rTCR-T cell therapy in solid tumors.

## 9. Incorporation of Suicide Genes to Safeguard Off-Target Toxicities

Mechanisms to eliminate the gene engineered T cell product can have utility in the case of on-target/off-tumor or oncogenic mutations events. Suicide genes have originally been used to modify T cells and decrease the onset of Graft versus Host Disease (GvHD). Integration of inducible suicide genes, resulting in cell death of the gene engineered T cells, allows a better control of side effect and give the possibility to eliminate the product post-infusion. A well-defined suicide gene is the herpes simplex virus thymidine kinase (HSV-tk), which has already been used to restrain graft versus host disease after applying ACT in a variety of malignancies. Activation of HSV-tk is achieved after applying ganciclovir (GCV), resulting in elimination of construct-engineered T cells. HSV-tk has already been included in some CAR T cell therapies, targeting the overexpressed protein CD44 isoform variant 6 (CD44v6), in clinical trials for the treatment of AML and MM patients (NCT04097301) [102]. Recently, the same CAR-T cells were able to reach, infiltrate, and proliferate at tumor sites in an adenocarcinoma tumor model [103]. In all those studies, effective elimination of the infused CAR T cells could be observed upon administration of GCV. Moreover, the use of HSV-tk is also beneficial for tracing the infused cells in the patients, being a valuable PET reporter gene. Using this imaging technique, in a mouse model of sarcoma, CAR-T cells were traced to the tumor, evaluating the efficiency of the therapy and the ablation upon GCV administration [104]. These results could be well translatable to rTCR-T cell therapies. However, there are some disadvantages related to the use of HSV-Tk genes, such as the immunogenicity of a viral derived gene-product, that can lead to immune activation and removal of therapeutic T cells; the amount of time needed before the T cells are eliminated by HSV-tk; and the need for the first line therapeutic agent for cytomegalovirus (CMV) infections [105]. The immunogenicity of a viral derived protein was overcome with the use of the inducible human caspase 9 (iCasp9) suicide gene. iCasp9 is a fusion protein of the catalytic domain of the pro-apoptotic protein caspase-9 and a domain of FKBP12 and can be activated after the introduction of a chemical inducer of dimerization (CID), such as AP1903 (Rimiducid) and AP20187. The CID can only bind the mutated domain of FKBP12 fused with iCASP9, but not the wild-type domain. Apoptosis of the infused T cells containing the construct is achieved after dimerization of the FKBP12 domains of iCasp9 upon binding of the CID, resulting in activation of the caspase molecules [106]. Robust elimination of donor T lymphocytes has already been shown after hematopoietic stem cell transplantation, underlying the great potential in rTCR-T cell therapy as well. More than 90% of T cells expressing the construct were eliminated within half an hour after delivery of AP20187, resulting in a diminished graft versus host reaction [107]. Further modification in the iCas9 system substitutes the use of AP1903 (Rimiducid) and AP20187 with Rapamycin. Rapamycin is involved in the inhibition of mTORC1, with FKBP12 as a co-factor, and is used as a well-tolerated immunosuppressive drug. Binding of rapamycin to FKBP12 increases the affinity for the unique FRB domain of TOR. It has been demonstrated that iRC9, a construct comprised of the catalytic domain of caspase 9 and both the FRB and FKBP12 domains, could functionally be activated by rapamycin, both in vitro and in vivo, in a mouse model of leukemia [108].

## 10. Conclusions

Over the past few decades, immunotherapy has made tremendous progress in improving cancer treatment options. In particular, patients suffering from hematological malignancies have gained clinical benefits, in some cases resulting in complete remission. However, solid tumors remain a greater challenge in this field to overcome. Genetic modification of T cells could address some of the current constraints. In this review, we focused on the use of rTCR-T cell therapy for solid tumors, from ongoing clinical trials to the future applications using cutting-edge genetic strategies to improve both potency and safety (Figure 3). To be efficacious and safe, rTCR-T cell products have to specifically and effectively target tumor cells and retain functionality within the suppressive tumor environment. Recently developed engineering techniques can help to address these limitations, by combining multiple gene addition approaches into a single treatment, such as gene addition of chemokine receptors and recombinant cytokines. Gene-editing techniques using CRISPR/Cas9 are highly efficient to knock out the endogenous TCR which significantly enhances the safety profile with respect to off-target peptide recognition, and to disrupt inhibitory signaling, which increase the persistency and functionality of the cells. In the context of rTCR-T cell therapy for solid tumors, the potential of gene augmentation and gene editing can be largely applied, especially to improve migration and infiltration into the tumor and T cell persistency after infusion. We discussed some of the possible strategies to target multiple pathways involved in the effectiveness of the T cell therapy, but we expect that many more approaches, targeting different pathways/signals, will be explored and might be beneficial to the final T cell product. It is important to move towards more specific personalized treatment regimens, in which multiplex combinations generate a tumor specific cell therapy product. We speculate that, in the future, combining deep, patient-individualized knowledge of solid tumors, with gene augmentation and gene editing techniques applied to rTCR-T cells, is the key to a successful and broadly applicable therapy. Moreover, it would be beneficial to have more accessibility to GMP-grade available components for gene-editing and to options of automation to standardize the production of multiplexed T cell therapy products in enclosed systems. Although gene augmentation and gene editing methods showed promising pre-clinical benefits, possible major side effects using these novel techniques should be addressed, but initial results indicate that the off-target gene-editing risk in T cells is low. Encouraging results have been presented in clinical trials in which cancer patients were infused with multiplexed genetically modified cells, proving initial safety of the drug product. 

In conclusion, the success of rTCR-T cells for solid tumor treatment could lie in the complexity of tools to genetically modify these cells on multiple levels creating a balance between efficacy and safety. The general use of these highly advanced rTCR-T cell therapies is coming closer to being a potential treatment option for solid tumors and is highly likely to be implemented in future T cell products for clinical application. 

## Figures and Tables

**Figure 1 vaccines-08-00733-f001:**
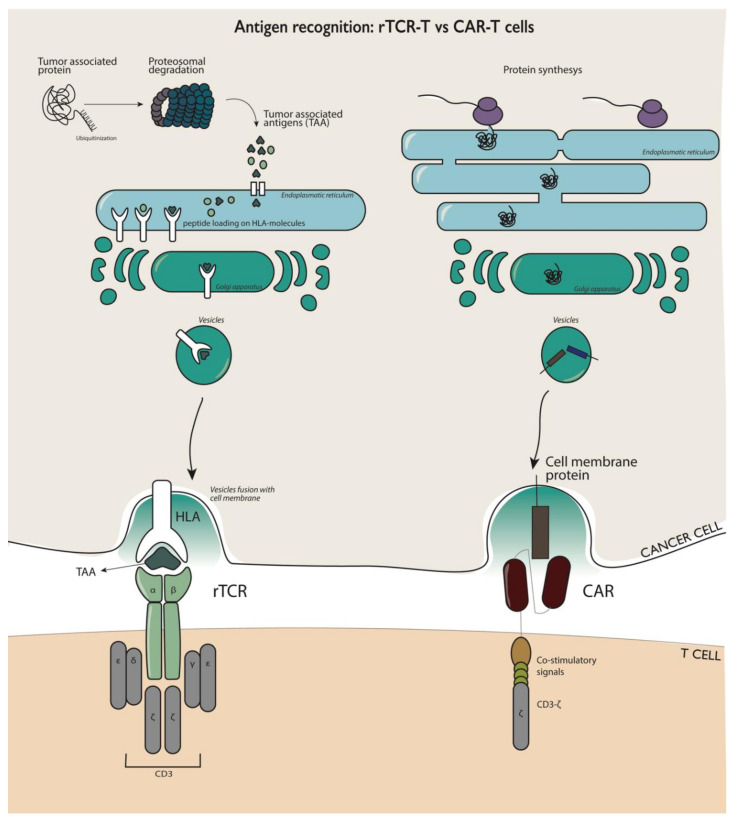
Differences in the mode of action of antigen recognition and activation of rTCR-T cells (on the left) and CAR-T cells (on the right). rTCR cells can recognize the presented peptide from both intracellular and extracellular proteins that are proteolytically processed and presented through human leukocyte antigen (HLA) molecules. CAR-T cells can solely recognize extracellular membrane proteins.

**Figure 2 vaccines-08-00733-f002:**
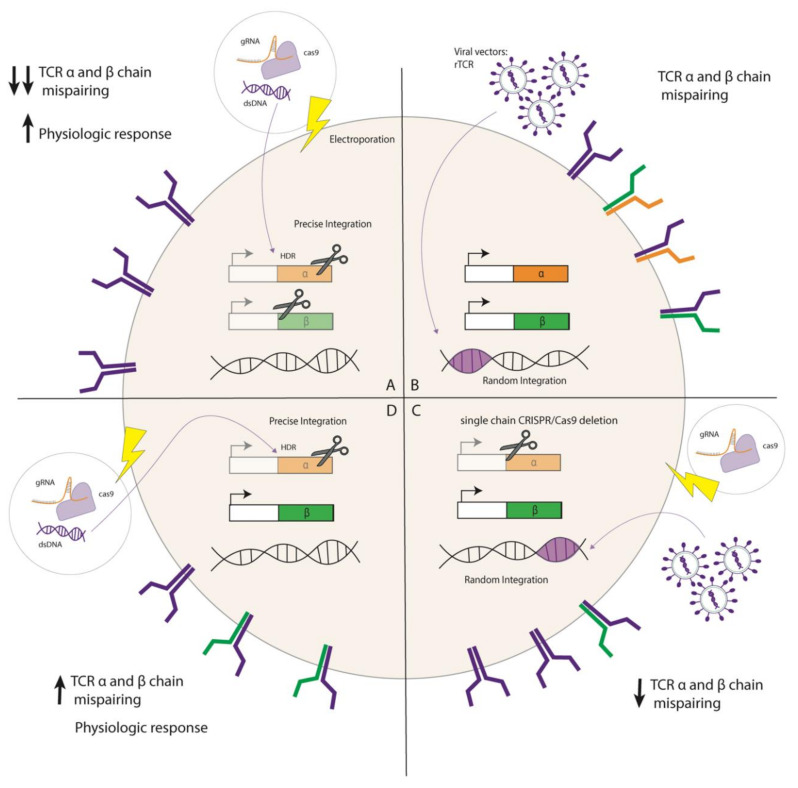
Options to combine gene-editing with rTCR gene augmentation, with precise or random integration. (**A**) Precise integration of rTCR in the *TRAC* locus with simultaneous disruption of *TRBC* locus decreases the possibility of TCR mispairing and is characterized by a physiological and endogenously controlled expression of the rTCR. KO only the *TRAC* locus when using random integration techniques decrease the chance of TCR molecule mispairing (**C**) compared to a not-gene editing approach (**B**). Notably knocking-in the rTCR into the *TRAC* locus increased the rate of TCR mispairing on the cell surface (**D**). (Schematic representation of data obtained by Schober et al. [55]).

**Figure 3 vaccines-08-00733-f003:**
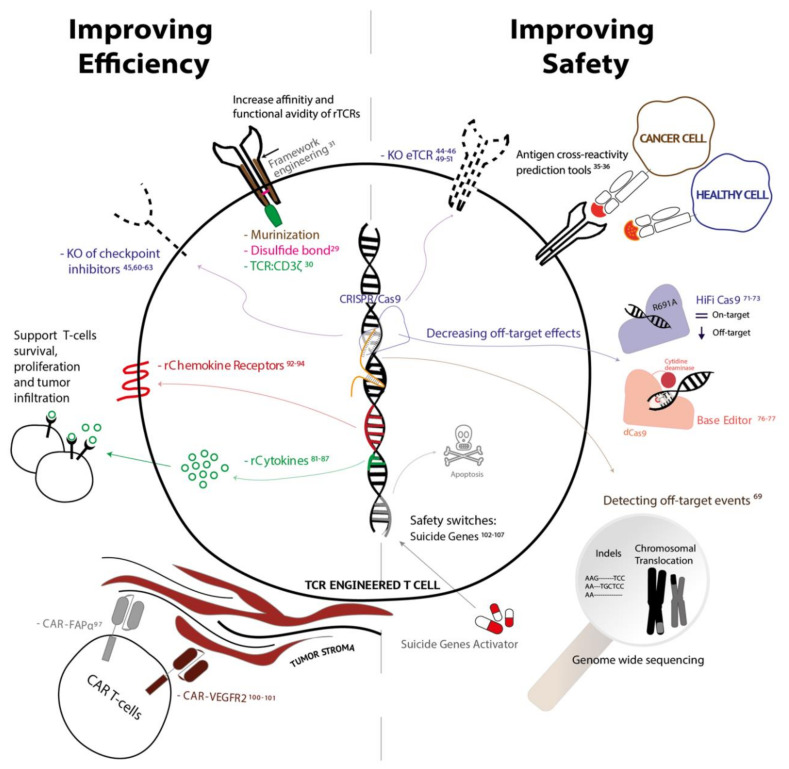
Summary of gene-augmentation and gene-editing strategies to improve efficiency (on the left) and safety (on the right) to treat solid tumors.

**Table 1 vaccines-08-00733-t001:** Registered clinical trial using rTCR-T cells against solid tumors (clinicaltrial.gov).

Target	Trial ID Number	Cancer Type	Additional Treatment	Location	Date	Status
NY-ESO-1	NCT01967823	Melanoma Meningioma Breast Cancer (and 2 more…)	Lymphodepletion Aldesleukin	USA	24 October 2013	Completed
	NCT02775292	Adult Solid Neoplasm Childhood Solid Neoplasm Metastatic Neoplasm	Lymphodepletion Aldesleukin Nivolumab	Unites States	3 January 2017	Completed
	NCT03029273	Lung Cancer, Non-small Cell, Recurrent	Lymphodepletion	China	21 March 2017	Recruiting
	NCT02650986	Advanced Fallopian Tube Carcinoma Advanced Malignant Solid Neoplasm Advanced Melanoma (and 47 more…)	Lymphodepletion TGF-β blocker	USA	30 June 2017	Recruiting
	NCT01567891	Ovarian Cancer	Lymphodepletion Aldesleukin	USA	3 July 2017	Completed
	NCT03017131	Recurrent Fallopian Tube Carcinoma Recurrent Ovarian Carcinoma Recurrent Primary Peritoneal Carcinoma	Lymphodepletion Aldesleukin	USA	8 December 2017	Recruiting
	NCT03638206	Multiple Myeloma Oesophagus Cancer Lung Cancer (and 13 more…)	Lymphodepletion	China	1 March 2018	Recruiting
	NCT03462316	Bone Sarcoma	Lymphodepletion Aldesleukin	China	21 May 2018	Recruiting
	NCT03709706	NSCLC	Pembrolizimab	USA	31December 2018	Recruiting
	NCT03691376	Platinum-Resistant Fallopian Tube Carcinoma Platinum-Resistant Ovarian Carcinoma Platinum-Resistant Primary Peritoneal Carcinoma (and 9 more…)	Chemotherapy Aldesleukin Cellular Therapy	USA	24 January 2019	Recruiting
	NCT03941626	Oesophagus Cancer Hepatoma Glioma Gastric Cancer	Lymphodepletion	China	1 September 2019	Recruiting
	NCT03967223	Neoplasms	Lymphodepletion	USA	31 December 2019	Recruiting
HPV E7	NCT02858310	Papillomavirus Infections Cervical Intraepithelial Neoplasia Carcinoma In Situ (and 2 more…)	Lymphodepletion	USA	27 January 2017	Recruiting
	NCT03912831	Human Papillomavirus (HPV) 16+ Relapsed/Refractory Cancer	Lymphodepletion	USA	8 June 2019	Recruiting
	NCT03937791	Squamous Intraepithelial Lesions of Vulva Neoplasms, Squamous Cell Vulvar HSIL	N/A	USA	9 October 2019	Recruiting
	NCT04411134	Cervical Intraepithelial Neoplasia	N/A	USA	5 June 2020	Not yet recruiting
	NCT04044950	Papillomavirus Infections Oropharyngeal Neoplasms	N/A	USA	5 June 2020	Not yet recruiting
	NCT04015336	Papillomavirus Infections Oropharyngeal Neoplasms	Lymphodepletion Aldesleukin	USA	5 June 2020	Recruiting
HPV E6	NCT02280811	Vaginal Cancer Cervical Cancer Anal Cancer (and 2 more…)	Lymphodepletion Aldesleukin	USA	14 October 2014	Completed
	NCT03197025	Human Papillomavirus HPV-16 High Grade Squamous Intraepithelial Lesion	Aldesleukin	USA	9 January 2018	Completed
	NCT03578406	Cervical Cancer Head and Neck Squamous Cell Carcinoma	PD-1 Antagonist	China	1 September 2018	Recruiting
	NCT04139057	Head and Neck Squamous Cell Carcinoma	PD-1 Antagonist	China	1 March 2019	Recruiting
MAGE family	NCT02111850	Cervical Cancer Renal Cancer Urothelial Cancer (and 2 more…)		USA	7 February 2014	Recruiting
	NCT03139370	Solid Tumor		USA	8 May 2017	Recruiting
	NCT03247309	Solid Tumor Cancer Head and Neck Squamous Cell Carcinoma Non-small Cell Lung Cancer		USA	19 December 2018	Recruiting
	NCT03441100	Solid Tumor, Adult Cancer Hepatocellular Carcinoma (and 4 more…)		USA Germany	2 May 2019	Recruiting
Neoantigens	NCT03412877	Glioblastoma Non-Small Cell Lung Cancer Ovarian Cancer (and 2 more…)		USA	6 September 2018	Recruiting
	NCT03970382	Solid Tumor		USA	3 July 2019	Recruiting
	NCT04102436	Glioblastoma Non-Small Cell Lung Cancer Breast Cancer (and 2 more…)		USA	5 June 2020	Recruiting
MART1 F5	NCT00509288	Melanoma Skin Cancer		USA	June 2007	Completed
	NCT00910650	Metastatic Melanoma		USA	13 October 2009	Completed
HBV antigen	NCT02719782	Recurrent Hepatocellular Carcinoma		China	2 July 2015	Recruiting
	NCT02686372	Hepatocellular Carcinoma		China	December 2015	Recruiting
EBV antigen	NCT03648697	Nasopharyngeal Carcinoma		China	10 October 2018	Not yet recruiting

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
