# Peer review of "Gene Augmentation and Editing to Improve TCR Engineered T Cell Therapy against Solid Tumors"

_vaccines, 2020, doi:10.3390/vaccines8040733_

Round 1

Reviewer 1 Report

The use of effective CAR-T cells and TCR engineered T cells have entered the clinic recently for the treatment of hematological malignancies with promising results. Hence, the topic of the present review is significant. The authors review the recent advance in the field and limitations that need to be overcome for applying these revolutionizing technologies to the treatment of malignant solid tumors. The contents are timely and informative and would be beneficial to general readers. However, there are many technical terms that are not explained clearly. All the unclear descriptions need to be clarified.   

  1. The authors described the structural details of TCR engineered T cells and CAR-T cells, but it is hard to follow in the current descriptions (e.g., HLA molecules increases the possibility to recognize a non-cognate peptide, derived from a different protein, the so-called off-target effect). Including an additional Figure regarding TCR and CAR-T would be helpful to clarify the explanations. 
  2. Line 130, cytokine-inducible SH2-containing protein (CISH), can negatively regulate depression by targeting PLC-gamma1 for degradation, an intermediate of the TCR. This needs an explanation. 
  3. MAGE-3-TCR should be explained.
  4. What is reducing the functionality of the T cell product?
  5. eTCR should be explained.
  6. Line 184, CRISPR-based genome editing result up to 90% in primary cells sounds too high. This point should be clarified. 
  7. An additional description of CD52 depletion should be added.
  8. Line 221, it is described that gene editing without the use of AAV would be safer due to the presence of pre-existing humoral responses in vivo against AAV. Generally, pre-existing neutralizing antibodies could deactivate the vector, not for the safety concern. An additional explanation on this point would be helpful. 
  9. Regarding reference #54, it is hard to follow the description without the guide of an image. 
  10. Line 310, Cas_OFFinder should be explained.
  11. Line 340, what does the genetically coupling mean?
  12. Line 349, Fix all the errors such as IFN-y
  13. Line 383, the reference #88 was cited for Magnetic Particle Imaging, but this study is not peer-reviewed yet.
  14. Related to reference #103, the recently published relevant article should be cited (PET reporter gene imaging and ganciclovir-mediated ablation of chimeric antigen receptor T-cells in solid tumors, Cancer Research, 2020).

Author Response

Reviewer 1

The use of effective CAR-T cells and TCR engineered T cells have entered the clinic recently for the treatment of hematological malignancies with promising results. Hence, the topic of the present review is significant. The authors review the recent advance in the field and limitations that need to be overcome for applying these revolutionizing technologies to the treatment of malignant solid tumors. The contents are timely and informative and would  beneficial to general readers. However, there are many technical terms that are not explained clearly.

Thank you for considering our review beneficial for the readers, and your suggestions to clarify technical terms more clearly for the reader.

All the unclear descriptions need to be clarified.   

  1. The authors described the structural details of TCR engineered T cells and CAR-T cells, but it is hard to follow in the current descriptions (e.g., HLA molecules increases the possibility to recognize a non-cognate peptide, derived from a different protein, the so-called off-target effect). Including an additional Figure regarding TCR and CAR-T would be helpful to clarify the explanations. 

Answer: An additional figure to visualize the differences between TCR engineered T cells and CAR-T cells has been included in the text.

  1. Line 130, cytokine-inducible SH2-containing protein (CISH), can negatively regulate depression by targeting PLC-gamma1 for degradation, an intermediate of the TCR. This needs an explanation. 

Answer: Sentence now at line 140 changed in:

“For instance, the cytokine-inducible SH2-containing (CISH) protein physically interacts with the TCR intermediate phospholipase Cγ1 (PLC-γ1), targeting it for proteasomal degradation after TCR stimulation by the cognate peptide, therefore decreasing the TCR expression on the surface.”

  1. MAGE-3-TCR should be explained.

Answer: MAGE-A3 stands for melanoma associated antigen A3, part of the melanoma antigen family. We included the full description in the text.

  1. What is reducing the functionality of the T cell product?

Answer: The functionality of T cell product is affected by several components, that are discussed throughout the manuscript. Here a summary of the main reasons why the T cell product decrease functionality and the location in the manuscript  

  1. The upregulation of checkpoint receptors decrease persistency after infusion (page 13 – Disrupting the inhibitory pathways to prevent exhaustion)
  2. After the infusion of the T cell product, the presence of inhibitory signals and the lack of stimulatory signals can also decrease the proliferation and functionality of the final product. (Page 16- Incorporating cytokines to enhance T cell proliferation; page 17- Introduction of chemokines receptors to promote migration and infiltration
  3. More specifically, in the context of solid tumors: the physical and chemical barrier of the tumor surrounding stroma and the suppressive microenvironment further decreases the migration of the T cell product and the ability to effectively eliminate tumor cells. (Page 19- Targeting the tumor surrounding stroma)

  1. eTCR should be explained.

Answer: this is an abbreviation of ‘endogenous TCR’. This has been mentioned in the text now.

  1. Line 184, CRISPR-based genome editing result up to 90% in primary cells sounds too high. This point should be clarified.

Answer: As reported in the research article of Ren et al. [ref 44] the use of CRISPR/Cas9 in primary T cells was succesfuly disrupting the target gene (the constant region of the TCR α-chain) with an efficiency of ~90%. Specifically, the mentioned reference is a chapter of the recent book “RNA Interference and CRISPR Technologies” that extensively summarizes the efficiency of CRISPR/Cas9 technology in primary T cells, compared to ZFN and TALENs.

  1. An additional description of CD52 depletion should be added.

Answer: Added in the text:

Line 206: “Additionally, to the beneficial effects of the eTCR disruption, the genetic disruption of CD52 expression protected the infused T cells from the depleting effect of alemtuzumab.”

  1. Line 221, it is described that gene editing without the use of AAV would be safer due to the presence of pre-existing humoral responses in vivo against AAV. Generally, pre-existing neutralizing antibodies could deactivate the vector, not for the safety concern. An additional explanation on this point would be helpful. 

Answer: The sentence was indeed referring to the pre-existing neutralizing antibodies that might decrease the efficiency of the therapy in case of recognition of residual immunogeneic viral proteins. Moreover, an extra sentence concerning the possibility to move faster into clinical applications with non-viral materials was added in the text:

“Line 241: to accelerate clinical application, considering that non-viral materials typically can be more easily adapted to good manufacturing practices for clinical use.”

  1. Regarding reference #54, it is hard to follow the description without the guide of an image.

Answer: Schematic image included in the text.

  1. Line 310, Cas_OFFinder should be explained.

Answer: We did not go in deep detail on the algorithm part but specified what the advantages are of using this prediction tool:

Line 331: “Using the prediction tool Cas-OFFinder, this approach is not limited to the variety in the protospacer-adjacent motif (PAM), mismatches and gRNA length, 2,086 potential off-target sites were predicted.”

  1. Line 340, what does the genetically coupling mean?

Answer: For genetically coupling it was intended to express recombinant IL2 and the recombinant TCR at the same time. However, to avoid confusion to the reader the sentence was modified in: “Moreover, expressing the IL-2 transgene simultaneously with the rTCR could further improve the engineered T cells with reduced systemic toxicity.”

  1. Line 349, Fix all the errors such as IFN-y.

Answer: This was adapted.

  1. Line 383, the reference #88 was cited for Magnetic Particle Imaging, but this study is not peer-reviewed yet.

Answer: That’s correct. However, we think it is a relevant study for the field but it will have to wait to be peer-reviewed. We focused on previous work using the biomedical imaging techniques.

  1. Related to reference #103, the recently published relevant article should be cited (PET reporter gene imaging and ganciclovir-mediated ablation of chimeric antigen receptor T-cells in solid tumors, Cancer Research, 2020).

Answer: Thank you for pointing to this reference. This was included in the review.

Reviewer 2 Report

The manuscript is well-written and provides valuable literature reviews. To improve the paper, some comments are listed below:

  1. The gene therapy did not discuss in the entire manuscript. Please consider modifying the Title “Gene therapy and editing to improve TCR engineered T cell therapy against solid tumors”.
  2. I suggest use “TCR-T” to replace “TCR engineered T” in the entire manuscript.
  3. The advantage of TCR-T cell therapy is not described well in the “Introduction section. Please consider using “the challenge of the TCR-T therapy” or “introduction of the challenge of the TCR-T therapy”
  4. There are no references for table 1.
  5. Page 7 “Increasing affinity and functional avidity of rTCRs while maintaining a safe profile”, This section mixes affinity of TCR and intrinsic inhibitory signal. I suggest the authors may need to separate into two sections.
  6. Line 117, “An additional limiting factor interfering with the functional avidity of TCRs is the expression on the surface.” Please described more clearly
  7. Some abbreviations did not explain, lease check the entire manuscript. i.e. eTCR, HA2, CDH13, EphA2
  8. Some typos, line 213, line 294, line 349 and 354
  9. Please provide authors’ view of points to improve the TCR-T cell therapy in the “Conclusion” section.

Author Response

Reviewer 2

The manuscript is well-written and provides valuable literature reviews. To improve the paper, some comments are listed below:

Thank you for considering our review well written and valuable for the field, and providing suggestions to improve the paper.

  1. The gene therapy did not discuss in the entire manuscript. Please consider modifying the Title “Gene therapy and editing to improve TCR engineered T cell therapy against solid tumors”.

Answer: We evaluated the suggestion and decided to change the title into “Gene augmentation and gene editing to improve TCR engineered T cell therapy against solid tumors”

  1. I suggest use “TCR-T” to replace “TCR engineered T” in the entire manuscript.

Answer: we replaced “TCR engineered T” with “rTCR-T”.

  1. The advantage of TCR-T cell therapy is not described well in the “Introduction section.

Please consider using “the challenge of the TCR-T therapy” or “introduction of the challenge of the TCR-T therapy”

Answer:We changed the title into:  “Advantages and disadvantages of using rTCR T cells for the treatment of solid tumor: from the bench to the bedside”.

  1. There are no references for table 1.

Answer: The table was generated using the clinicaltrial.gov website, information that was added to the legend. Unfortunately not all the registered trial have reported and published results yet.

  1. Page 7 “Increasing affinity and functional avidity of rTCRs while maintaining a safe profile”, This section mixes affinity of TCR and intrinsic inhibitory signal. I suggest the authors may need to separate into two sections.

Answer: We prefer to keep those two parts together. The intrinsic inhibitory signaling (CISH/PLC-γ1) is mentioned as a possible strategy to enhance the functional avidity of TCR engineered T cells, as expressed by the title.

  1. Line 117, “An additional limiting factor interfering with the functional avidity of TCRs is the expression on the surface.” Please described more clearly

Answer: The text was adapted to describe this more clearly: “Genetic modification of key amino acids, identified with the use of phage display and point mutations, has shown the possibility to further increase the TCR affinity, evaluated measuring the binding of the rTCR to the HLA-bound cognate peptides [24,25].  If the TCR affinity is important at the level of antigen recognition, TCR functional avidity is essential to obtain an actual cytotoxic effect. Enhancing the rTCR expression on the cell surface is one of the strategies to increase the functional avidity. High level of rTCR expressed on the cell surface can double the production of cytotoxic cytokines, such of IFNγ, compared to a low expression [26].

  1. Some abbreviations did not explain, lease check the entire manuscript. i.e. eTCR, HA2, CDH13, EphA2.

Answer: The full names of those abbreviation have been included in the text.

  1. Some typos, line 213, line 294, line 349 and 354.

Answer: These have been adapted.

  1. Please provide authors’ view of points to improve the TCR-T cell therapy in the “Conclusion” section.

Answer: The conclusion section was implemented with the authors’ point of view.

Reviewer 3 Report

The manuscript discuss the use of gene engineered T cells receptors in the treatment of cancerous solid tumors. It is a deep and updated review article, covering extensive literature (107 references) on the subject. A concluding summary illustrated in Figure 1, proposes a very convincing conclusion. It mention the need for a well balance between improved efficiency and safety in gene-therapy and gene-editing strategies that could be adopted to treat solid tumors. Thus, the work is clearly useful to subsidize decisive clinical applications about the treatment of several types of cancerous tumors. So, I didn’t find any justification contrary to the recommendation of publication of the manuscript.

Author Response

We would like to thank reviewer 3 for the statements regarding our manuscript and for deeply understanding the purpose and final goals of the manuscript.

Round 2

Reviewer 1 Report

The authors responded appropriately to this reviewer's comments. A few typographic errors need to be corrected.